# The Optimizing Background Infusion Mode Decreases Intravenous Patient-Controlled Analgesic Volume and Opioid Consumption Compared to Fixed-Rate Background Infusion in Patients Undergoing Laparoscopic Cholecystectomy: A Prospective, Randomized, Controlled, Double-Blind Study

**DOI:** 10.3390/medicina57010042

**Published:** 2021-01-06

**Authors:** Ki Tae Jung, Keum Young So, Seung Un Kim, Sang Hun Kim

**Affiliations:** 1Department of Anesthesiology and Pain Medicine, School of Medicine, Chosun University, 309 Pilmun-daero, Dong-gu, Gwangju 61452, Korea; mdmole@chosun.ac.kr (K.T.J.); kyso@chosun.ac.kr (K.Y.S.); 2Department of Anesthesiology and Pain Medicine, Chosun University Hospital, 365 Pilmun-daero, Dong-gu, Gwangju 61453, Korea; sekizang@naver.com

**Keywords:** background infusion, intravenous infusions, laparoscopic cholecystectomy, opioid analgesics, patient-controlled analgesia, postoperative pain

## Abstract

*Background and objectives:* The fixed-rate continuous background infusion mode with bolus dosing is a common modality for intravenous patient-controlled analgesia (PCA). However, some patients suffer from inadequate analgesia or opioid-related adverse effects due to the biphasic pattern of postoperative pain. Therefore, we investigated the postoperative analgesic efficacy of PCA using an optimizing background infusion mode (OBIM) where the background injection rate varies depending on the patient’s bolus demand. *Materials and Methods:* We prospectively enrolled 204 patients who underwent laparoscopic cholecystectomy in a randomized, controlled, double-blind study. Patients were allocated to either the optimizing (group OBIM) or the traditional background infusion group (group TBIM). The numeric rating scale (NRS) score for pain was evaluated at admission to and discharge from the recovery room, as well as at the 6th, 24th, and 48th postoperative hours. Data on bolus demand count, total infused volume, and background infusion rate were downloaded from the PCA device at 30-min intervals until the 48th postoperative hour. *Results:* The NRS score was not significantly different between groups throughout the postoperative period (*p* = 0.621), decreasing with time in both groups (*p* < 0.001). The bolus demand count was not significantly different between groups throughout (*p* = 0.756). The mean total cumulative infused PCA volume was lower in group OBIM (84.0 (95% confidence interval: 78.9−89.1) mL) than in group TBIM (102 (97.8−106.0) mL; *p* < 0.001). The total cumulative opioid dose in fentanyl equivalents, after converting sufentanil to fentanyl using an equipotential dose ratio, was lower in group OBIM (714.1 (647.4−780.9) μg) than in group TBIM (963.7 (870.5−1056.9) μg); *p* < 0.001). The background infusion rate was significantly different between groups throughout the study period (*p* < 0.001); it was higher in group OBIM than in group TBIM before the 12th postoperative hour and lower from the 18th to the 48th postoperative hour. *Conclusions:* The OBIM combined with bolus dosing reduces the cumulative PCA volume and opioid consumption compared to the TBIM combined with bolus dosing, while yielding comparable postoperative analgesia and bolus demand in patients undergoing laparoscopic cholecystectomy.

## 1. Introduction

Intravenous patient-controlled analgesia (PCA) is a common method of immediately delivering analgesics on an as-required basis to the patient via an infusion pump [1]. Its main benefit is the provision of appropriate analgesia according to patient demand, ultimately increasing patient satisfaction [2,3]. The most common PCA modes are intermittent, fixed demand dosing (self-administering) with or without continuous background infusion for postoperative analgesia [4,5]. Other variable parameters associated with PCA include the loading dose, bolus dose, lockout interval (time lag between bolus doses), and continuous background infusion rate [1].

However, despite using PCA devices, some patients experience inadequate analgesia due to the biphasic pattern of postoperative pain; it is more intense than anticipated immediately after surgery and less intense from the day after surgery [6,7]. Hence, patients may suffer from insufficient analgesia immediately after surgery, and may ultimately require frequent additional rescue analgesics because of the lockout interval and the fixed rate of a continuous background infusion [6,7]. They may also experience postoperative opioid-related adverse effects due to the combination of a self-administered bolus and fixed-rate continuous background infusion [8,9].

To address these shortcomings, the PAINSTOP medicine-injection pump (PS-1000, Unimedics Co., Ltd., Seoul, Korea) was introduced as a new PCA device providing an “optimizing background infusion mode” (OBIM) defined by the manufacturer [10]. The OBIM, also termed the “variable-rate feedback infusion mode” (VFIM), refers to the background injection rate that varies depending on bolus demand over a predefined time [1]. However, this mode is yet to be applied in clinical practice for postoperative pain control, and there is a lack of evidence of its utility [1,10,11].

We hypothesized that the OBIM would provide better postoperative analgesia and a lower cumulative opioid consumption compared to the traditional (fixed-rate) background infusion mode (TBIM). To investigate this, we evaluated the efficacy of PCA in patients undergoing laparoscopic cholecystectomy using bolus dosing and either the OBIM or the TBIM for postoperative analgesia. The primary outcome of this study was achievement of a medium effect size (0.5) in the numeric rating scale (NRS) pain score six-hours postoperatively.

## 2. Materials and Methods

### 2.1. Study Design and Ethical Statement

This prospective, randomized, controlled, double-blind study was approved by the Institutional Review Board of Chosun University Hospital (Chosun 2018-02-011) on 6 March 2018, and was prospectively registered with the Clinical Research Information Service (CRIS: https://cris.nih.go.kr/, ref: KCT0002777) on 5 April 2018. It was conducted according to the Declaration of Helsinki of 1964 and all of its subsequent revisions.

### 2.2. Selection of Study Population

The subjects included patients aged 20 to 70 years with an American Society of Anesthesiologists (ASA) physical status of I–III who were scheduled to undergo elective laparoscopic cholecystectomy under general anesthesia between 3 September 2018 and 14 February 2020. Written informed consent was obtained from all participants after a thorough explanation of the purpose of the study. Participants were instructed to push the “demand” button of the PAINSTOP device whenever they experienced pain of >4 points on the numeric rating scale (NRS: 0 = no pain, 10 = worst pain imaginable). We excluded patients with severe cardiopulmonary disease, renal or hepatic functional abnormalities, neuromuscular disorders, or a history of opioid-related complications.

### 2.3. Randomization and Masking

Two hundred four patients were randomly assigned to two groups that used a PCA device applying either the fixed-rate background infusion mode (group TBIM, *n* = 102) or the optimizing background infusion mode (group OBIM, *n* = 102). In addition, the enrolled patients in each of these two groups were assigned randomly to one of two further groups according to whether they received fentanyl or sufentanil. Randomization was performed using a computer-generated table of random numbers via the permuted block method (a 1:1 allocation ratio and a block size of 2). This randomization was performed using PASS 15 Power Analysis and Sample Size Software (2017) (NCSS, LLC., Kaysville, UT, USA).

The researcher who managed the anesthesia (RA) was responsible for obtaining informed consent from participants, as well as for gathering and recording data from the participants and the PCA devices. The researcher who managed the PCA (RP) was responsible for assigning the correct drugs to each PCA device according to the randomization scheme. For blinding, the RP recorded the drug assignment in the anesthesia charts after the anesthesia was finished, and the RA finally collated the data of patient medical records that were generated through the trial for at least 48 h postoperatively. The nurses in the recovery room (RR) or ward recorded postoperative pain and postoperative nausea and vomiting (PONV) using the NRS; these nurses were not part of the investigating team and were trained in the hospital to assess pain intensity and PONV with the NRS. Neither RA nor RP participated in the statistical analysis.

### 2.4. Interventions

After premedication with intramuscular midazolam (0.05 mg/kg), the patients were transported to an operating room. AA anesthetized the patients using total intravenous anesthesia with propofol and remifentanil and maintained the optimal neuromuscular paralysis with rocuronium. Ten minutes before the end of surgery, RP commenced the PCA device according to the group allocation, after administration of an initial bolus dose (2 mL; fentanyl: 0.29 μg/kg or sufentanil 0.04 µg/kg) from the PCA device and ramosetron (0.3 mg). We used an equianalgesic dose of sufentanil and fentanyl (fentanyl:sufentanil = 6:1) [12].

The total PCA volume was 140 mL, comprised of normal saline, fentanyl (20 μg/kg) or sufentanil (3 μg/kg), nefopam (160 mg), and ramosetron (1.2 mg). All PCA devices were initially set to administer a bolus of 2 mL (fentanyl: 0.29 μg/kg or sufentanil: 0.04 μg/kg) with a lockout interval of 10 min and a background infusion rate of 2 mL/h. The background infusion rate of group OBIM was set to increase automatically by 0.4 mL/h (fentanyl: 0.06 μg/kg/h or sufentanil: 0.01 μg/kg/h) each time a bolus dose was required, and decrease by 0.2 mL/h (fentanyl: 0.029 μg/kg/h or sufentanil: 0.004 μg/kg/h) when a bolus dose was not required for 1.5 h. The background infusion rate was limited to a maximum of 4.0 mL/h (fentanyl: 0.57 μg/kg/h or sufentanil: 0.09 μg/kg/h) and a minimum of 1 mL/h (fentanyl: 0.14 μg/kg/h or sufentanil: 0.02 μg/kg/h). All drug doses were based on the ideal body weight of patients. By comprehensively incorporating the opinions of manufacturers and researchers, due to lack of evidence for specific OBIM settings, we created final settings to ensure patient safety and to provide effective analgesia.

At the end of surgery, the patients did not receive any wound anesthetic infiltration with local anesthetics or any regional analgesia. The patients were transferred to the RR after the complete reversal of rocuronium-induced neuromuscular paralysis and when they were fully awake. When patients experienced pain in the RR of >4 points on the NRS, the RR nurse or the patient pushed the PCA button for administration of a bolus dose. When patients required additional rescue analgesics within the lockout interval, the RR nurse intravenously administered either ketorolac (30 mg) or nefopam (20 mg). We also allowed the intravenous injection of opioids, nonsteroidal anti-inflammatory drugs, or tramadol as rescue analgesics in the ward to treat pain of >4 points on the NRS. We treated PONV of >4 points on the NRS with the intravenous injection of metoclopramide (10 mg). Our research staff decided whether to stop the PCA device or change its settings based on the severity of signs and symptoms, and we excluded cases where this occurred from the final statistical analysis.

### 2.5. Outcomes

We recorded an NRS score for pain; PONV; and the need for additional rescue analgesics and antiemetics on admission to (RR1) and discharge from (RR2) the RR, as well as at the 6th, 24th, and 48th postoperative hours. We downloaded the data from the PCA device (bolus demand count, total infused volume, background infusion rate), using its built-in Wi-Fi system, in 30-min intervals until the 48th postoperative hour. We recorded data on demographics (age, sex, height, weight, ASA physical status, intraoperative remifentanil dose, operating time, anesthesia time, PCA composition) and perioperative complications, as well as the incidence of and causes for early termination of the PCA device.

### 2.6. Sample Size

To estimate the sample size for the primary outcome, we used G*Power software (ver. 3.1.9.1, Heinrich-Heine-Universität, Düsseldorf, Germany). We set the two-tailed level of statistical significance as α = 0.05, the power as 90%, and the medium effect size as 0.5 (defined by Cohen for analyses using the Student *t*-test); the latter was an assumption, as there were no previous data from which to calculate the effect size [13].

The study required 172 patients in total; thus, we enrolled 204 patients, allowing for a dropout rate of approximately 15%.

### 2.7. Analysis

IBM SPSS Statistics for Windows, ver. 26.0 (IBM Corp., Armonk, NY, USA) was used for all statistical analyses. All data were analyzed as if their probability distributions were normal based on the central limit theorem and are presented as means (95% confidence intervals (CI)), means ± standard deviation (SD), numbers (of patients (*n*), or numbers (percentage) of patients (*n* [%])). We analyzed continuous variables using the Student *t*-test and nominal variables with the χ^2^ or Fisher’s exact test. For analysis of time-interval data that passed Mauchly’s sphericity test, we used repeated measures analysis of variance; for data that did not pass Mauchly’s sphericity test, we used Wilk’s lambda multivariate analysis of variance. To compare two groups in a given time interval, the Student *t*-test was used. *p* values < 0.05 were considered statistically significant.

## 3. Results

### 3.1. Demographic Data

There were no important harms or unintended effects in either group in this study. We enrolled 204 patients finally; however, 71 patients were excluded from the final analysis, representing a 34.8% dropout rate (Table 1, Figure 1).

The number of excluded patients was significantly different between the groups (*p* < 0.001): 23 (22.5%) in group TBIM and 48 (47.1%) in group OBIM (Table 1). The reasons for exclusion included data loss during collection in 38 patients (18.6%), early PCA termination in 20 patients (9.8%), and device setting errors in 13 patients (6.4%; Table 1).

The causes of early PCA termination were postoperative nausea (two patients in group OBIM) and patient request due to a lack of pain (seven in group TBIM, 11 in group OBIM). However, the number of early PCA terminations was not significantly different between the groups (*p* = 0.214; Table 1).

No statistically significant differences were observed in demographic data, intraoperative variables, or PCA regimens after exclusion of the above patients (Table 2 and Table 3). The opioids (fentanyl and sufentanil) used in the PCA devices were not significantly different between the groups (fentanyl: 44 patients (55.7%) and 23 patients (42.6%) in groups TBIM and OBIM, respectively; *p* = 0.160).

### 3.2. NRS Scores

The NRS score was not significantly different between the groups throughout the postoperative period (*p* = 0.621), and it decreased with time in both groups (*p* < 0.001, Figure 2).

### 3.3. Bolus Demand Counts

The bolus demand count was not significantly different between groups throughout the postoperative period (*p* = 0.756, Figure 3).

### 3.4. Background Infusion Rate 

The background infusion rate was significantly different between groups throughout the postoperative period (*p* < 0.001, Figure 4a,b). The background infusion rate of group OBIM was significantly different from that of group TBIM for all time intervals except for the 12th postoperative hour (*p* < 0.001, Figure 4b). The background infusion rate was higher in group OBIM than in group TBIM before the 12th postoperative hour and lower from the 18th to the 48th hours (Figure 4b). The maximum and minimum background infusion rates were 3.3 (3.2−3.5) and 1.1 (1.0−1.2) mL/h, respectively in group OBIM, while the background infusion rate in group TBIM was constant at 2.0 mL/h.

### 3.5. Infused PCA Volumes and Infused Opioid Doses 

The cumulative infused PCA volume was significantly different throughout the postoperative period (*p* < 0.001) and at each measured interval (*p* ≤ 0.005) except at the 24th and 30th postoperative hours (Figure 5a). It was higher in group OBIM than in group TBIM until the 18th postoperative hour and lower from the 38th to the 48th postoperative hour (Figure 5a). The total cumulative infused volume was lower in group OBIM (84.0 (78.9−89.1) mL) than in group TBIM (102 (97.8−106.0) mL; mean difference (95% CI): 17.9 (11.6 to 24.2), *p* < 0.001, Figure 5a). The per-interval infused PCA volume was significantly different between groups throughout the postoperative period (*p* < 0.001, Figure 5b); it was higher in group OBIM than in group TBIM until the 12th postoperative hour and lower from the 24th to the 48th hours (*p* ≤ 0.004, Figure 5b).

In addition, we analyzed differences of the cumulative and per-interval infused opioid doses between the two groups after the sufentanil doses were converted to opioid doses in fentanyl equivalents. The cumulative infused opioid dose was significantly different throughout the postoperative period (*p* < 0.001) and at each measured interval (*p* ≤ 0.006) except at the 18th, 24th and 30th postoperative hours (Figure 6a). It was higher in group OBIM than in group TBIM until the 12th postoperative hour and lower from the 38th to the 48th postoperative hour (Figure 6a). The total cumulative opioid dose was lower in group OBIM (714.1 (647.4−780.9) μg) than in group TBIM (963.7 (870.5−1056.9) μg; mean difference (95% CI): 249.6 (133.5 to 365.6); *p* < 0.001, Figure 6a). The per-interval infused opioid dose was significantly different between groups throughout the postoperative period (*p* < 0.001, Figure 6b). It was higher in group OBIM than in group TBIM until the 6th postoperative hour and lower from the 18th to the 48th hours (*p* ≤ 0.008, Figure 6b).

### 3.6. Rescue Drugs and Complications

The proportion of patients requiring rescue analgesics and antiemetics was not significantly different between the groups throughout the recovery period (*p* = 0.165 and *p* = 0.686, respectively; Table 4). The specific postoperative rescue analgesics used were tramadol, diclofenac, and fentanyl, which were not significantly different between the groups throughout the recovery period (*p* ≥ 0.05, Table 5). The total cumulative opioid dose after the tramadol doses were converted to opioid doses in fentanyl equivalents was not significantly different between the groups (mean: 31.5 μg and 20.7 μg in groups TBIM and OBIM, respectively, and mean difference (95% CI): 10.8 (−11.4 to 33.0); *p* = 0.339, Table 5). The total cumulative diclofenac dose was not significantly different between the groups (mean: 1.1 mg and 1.7 mg in groups TBIM and OBIM, respectively, and mean difference (95% CI): −0.5 (−4.4 to 3.3); *p* = 0.787, Table 5).

## 4. Discussion

This prospective, double-blind, randomized controlled study revealed that the NRS score and bolus demand count did not differ between groups throughout the recovery period. Patients in group OBIM exhibited a higher background infusion rate before the 12th postoperative hour and a lower rate from the 12th to the 48th postoperative hours compared with those in group TBIM. OBIM offered adequate response to the biphasic pain pattern after laparoscopic surgery compared to TBIM. The total cumulative infused PCA volume and opioid consumption were lower in group OBIM than in group TBIM.

Many previous studies of PCA using the VFIM were conducted in patients using “computer-integrated” patient-controlled epidural analgesia (PCEA) during labor and delivery [14,15,16,17]. Their results suggested that patient satisfaction was greater in those using the computer-integrated PCEA than in those using traditional PCEA, but that the incidence of breakthrough pain and the cumulative local anesthetic consumption did not differ significantly between groups [15,17]. However, we are aware of only one other study in which the effect of intravenous PCA was evaluated using a similar VFIM technique to that of our study in combination with demand dosing, which was performed in patients undergoing spinal surgery [10]. In that study, the VFIM did not significantly decrease the NRS score for postoperative pain compared with the TBIM, and the NRS score decreased over time in both groups [10]. The cumulative infused PCA volume was significantly lower in the VFIM than in the TBIM group at the 24th and 48th postoperative hours. The authors assumed that this resulted from the corresponding lower bolus demand counts throughout the recovery period, with significantly lower bolus demand counts in the VFIM than in the TBIM group at the 12th and 24th postoperative hours [10]. Hence, they suggested that the VFIM could provide more efficient postoperative analgesia and reduce the cumulative infused PCA volume than the TBIM [10]. This study also demonstrated that the OBIM contributed to a reduced cumulative infused PCA volume during the first 48 postoperative hours. However, we observed no significant differences in NRS scores or bolus demand counts between the OBIM and TBIM groups. This may be explained by the relatively high proportion of patients receiving additional rescue analgesics throughout the recovery period in group TBIM, and by the relatively low pain following laparoscopic cholecystectomy compared with that following spinal surgery. If we restricted the use of additional rescue analgesics and studied patients who underwent more painful surgeries, the results may have differed.

Most patients experience a biphasic pattern of pain intensity that requires more analgesics immediately after surgery followed by less analgesics subsequently [6,7]. The fixed background infusion rate (TBIM) combined with bolus dosing is not enough to overcome early postoperative pain, because the lockout interval limits the rescue analgesia infusion via PCA despite frequent bolus demands [6,7,18]. However, the OBIM with bolus dosing has the benefit of providing more effective postoperative pain and reducing the total infused volume compared to the TBIM with bolus dosing [10,18], because the OBIM increases the background infusion rate for the increased bolus demands immediately after surgery and then decreases the background infusion rate due to the absence of a bolus-dosing requirement [10]. We have commonly adopted the PCA by using bolus dosing combined with the TBIM. However, the TBIM was associated with a higher incidence of insufficient postoperative analgesia and higher postoperative pain, resulting in an increase in total opioid consumption and higher postoperative bolus requirements compared to the OBIM [18]. Our study also showed that the OBIM reduced the total infused volume (total opioid consumption) even in the presence of non-significantly reduced postoperative NRS score associated with improved postoperative analgesia. So, PCA using the OBIM combined with bolus dosing can provide more sufficient postoperative analgesia and further reduce opioid consumption by decreasing the total infused volume.

Considering the biphasic postoperative-pain pattern, opioid-related adverse effects are a major concern in patients using PCA. The OBIM PCA may result in adverse effects because of an increased background infusion rate and an increased bolus demand due to high levels of pain experienced immediately after surgery. On the other hand, the TBIM PCA may result in an unnecessary infusion of opioids in patients that do not require active pain control beyond the acute period of postoperative pain [10]. This study revealed that postoperative nausea requiring antiemetics mainly occurred before the sixth postoperative hour in the OBIM group (3.7%), and after the sixth postoperative hour in the TBIM group (3.8%; Table 4). No other adverse effects were observed.

Lee at al. [10] documented that the overall incidence of PONV requiring antiemetics was lower in the OBIM group (18%) compared to the TBIM group (33%); in contrast, in this study it was higher in the OBIM group (5.6%) than the TBIM group (3.8%; Table 4). This discrepancy has several possible explanations. First, Lee at al. [10] used PCA devices with opioids alone, while we combined opioids and antiemetics. Our use of premixed antiemetics probably contributed to reducing the overall incidence of PONV in both groups compared to the study by Lee et al. [10]. Second, we did not confirm whether the PONV was directly related to the administered opioid dose as we did not record the incidence of PONV at each time interval. In this study, we enrolled patients who underwent laparoscopic surgery, an important risk factor for PONV. Even though the premixed antiemetics reduced the overall incidence of PONV, the risk of PONV was probably increased by the increased background infusion rate of opioids during the acute period in the OBIM group, resulting in a higher incidence of PONV in this group than in the TBIM group.

The major limitation of this study was the drop-out rate (34.8%), which was much higher than expected (15%). The causes included data loss when downloading from the PCA device (18.2%), early termination of the PCA (9.8%), and device-setting errors (6.4%). First, even though we allowed ample time for the RP to be trained in the setup of the PCA, we had to exclude 2.9% of patients in group TBIM and 9.8% in group OBIM due to setting errors. OBIM requires, in addition to the setup of the TBIM, the setup of conditions and sizes for increases and decreases in the background infusion rate as well as the maximum and minimum allowable background infusion rates. This complex setup, combined with unfamiliarity with the OBIM of the PCA device, requires ample training time to prepare and operate the device in order to reduce setup and operation errors [10]. Second, some patients who underwent laparoscopic surgeries were discharged early due to low levels of postoperative pain and a quick recovery. Third, part of the PCA data were lost as we overlooked the fact that the data is erased when the device is powered down. Therefore, a more secure system should be implemented for downloading data from the device in future studies.

Another limitation of this study is that we did not adopt multimodal pain management protocols including preemptive analgesia, non-steroidal anti-inflammatory drugs, gabapentinoids, acetaminophen, muscle relaxants, ketamine, neuro-axial blockade, and local infiltrative anesthetic. [19]. Therefore, we cannot conclude that OBIM will be able to provide more effective postoperative analgesia in the context of applying a multimodal pain management protocol with PCA.

## 5. Conclusions

The OBIM reduces cumulative PCA volume and opioid consumption by responding more effectively to postoperative pain compared to the TBIM, while yielding comparable postoperative analgesia and bolus demand in patients undergoing laparoscopic cholecystectomy. Further studies are required to determine the efficacy of the OBIM in different types of surgery and degrees of postoperative pain.

## Figures and Tables

**Figure 1 medicina-57-00042-f001:**
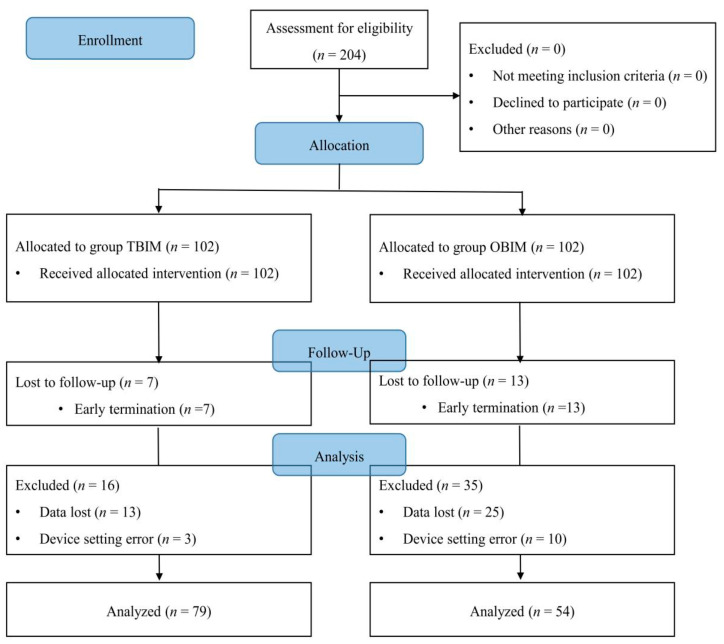
CONSORT diagram for patient recruitment. OBIM: optimizing background infusion mode, TBIM: traditional background infusion mode.

**Figure 2 medicina-57-00042-f002:**
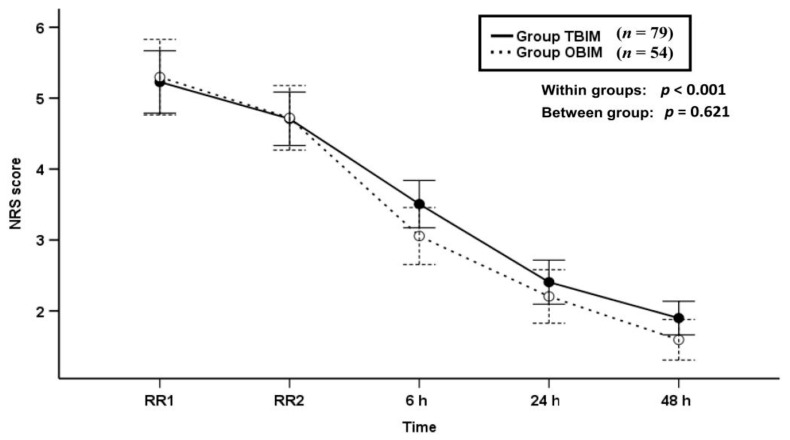
Time-sequential changes of numeric rating scale (NRS) scores. Data points and error bars represent means and 95% confidence intervals, respectively. OBIM: optimizing background infusion mode, TBIM: traditional background infusion mode, RR1: at admission from the recovery room, RR2: at discharge from the recovery room.

**Figure 3 medicina-57-00042-f003:**
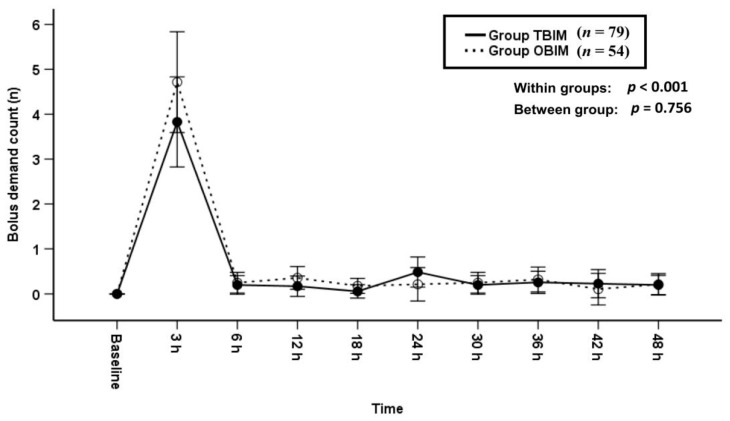
Time-sequential changes of bolus demand counts. Data points and error bars represent means and 95% confidence intervals, respectively. OBIM: optimizing background infusion mode, TBIM: traditional background infusion mode.

**Figure 4 medicina-57-00042-f004:**
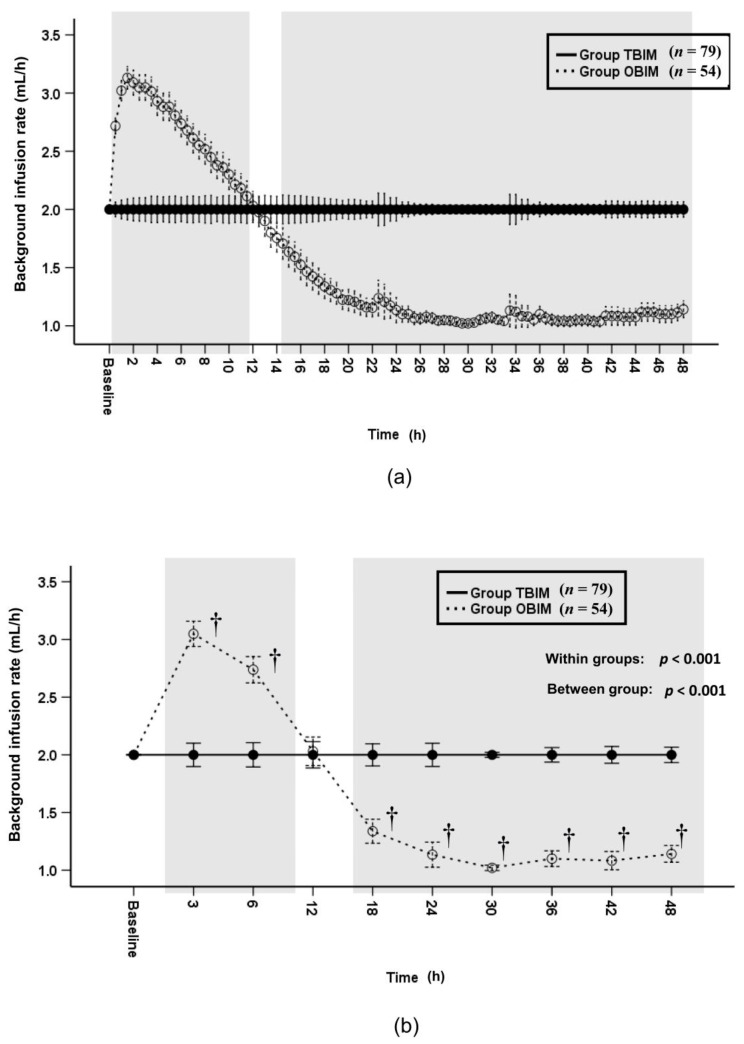
Time-sequential changes of background infusion rate at 30-min intervals (**a**) and at specific time points (**b**). Data points and error bars represent means and 95% confidence intervals, respectively. OBIM: optimizing background infusion mode, TBIM: traditional background infusion mode. †: *p* < 0.001 compared with group TBIM.

**Figure 5 medicina-57-00042-f005:**
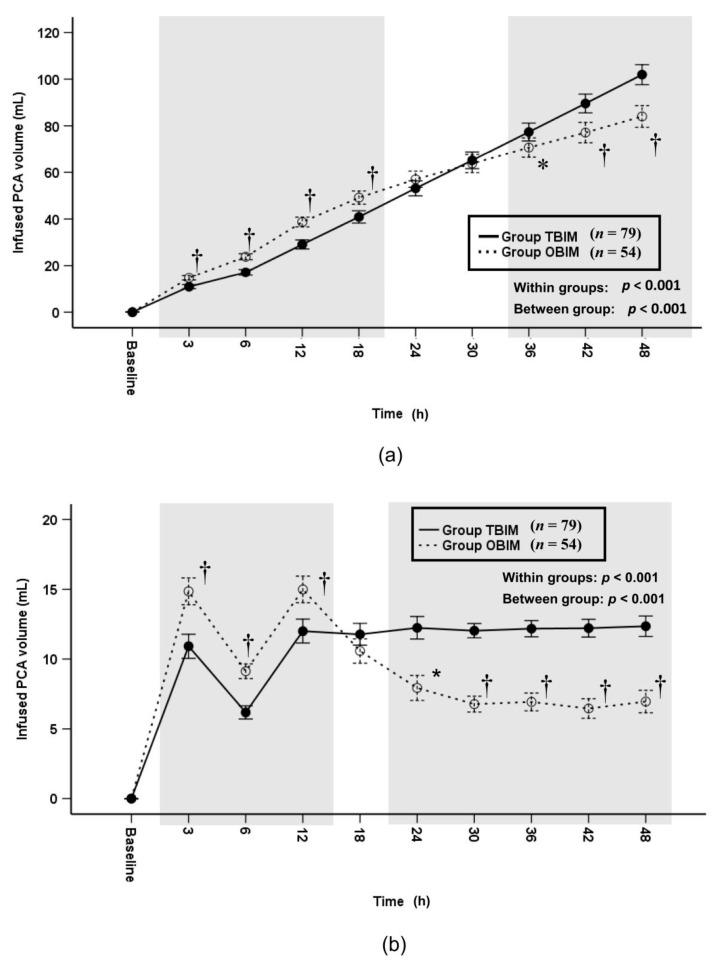
Time-sequential changes of cumulative (**a**) and per-interval (**b**) infused PCA volumes. The gray boxes represent the intervals in which there were statistically significant differences between the groups. Data points and error bars represent means and 95% confidence intervals, respectively. OBIM: optimizing background infusion mode, PCA: patient-controlled analgesia, TBIM: traditional background infusion mode. *: *p* < 0.05 compared with group TBIM, †: *p* < 0.001 compared with group TBIM.

**Figure 6 medicina-57-00042-f006:**
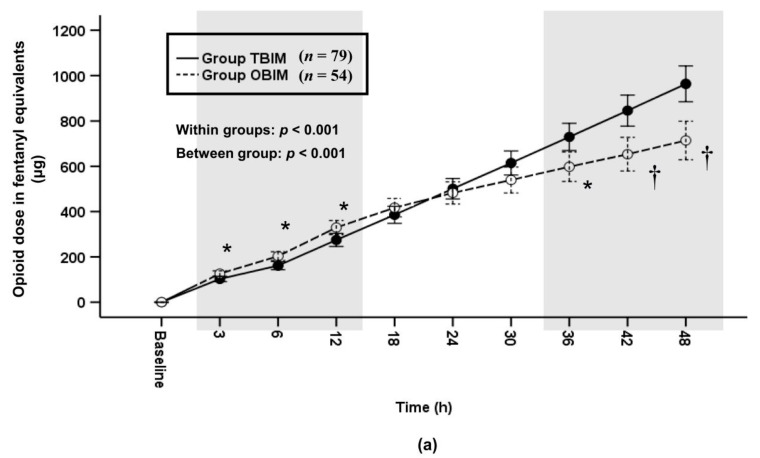
Time-sequential changes of cumulative (**a**) and per-interval (**b**) infused opioid dose in fentanyl equivalents. The gray boxes represent the intervals in which there were statistically significant differences between the groups. Data points and error bars represent means and 95% confidence intervals, respectively. OBIM: optimizing background infusion mode, PCA: patient-controlled analgesia, TBIM: traditional background infusion mode. *: *p* < 0.05 compared with group TBIM, †: *p* < 0.001 compared with group TBIM.

**Table 1 medicina-57-00042-t001:** Incidences and causes for exclusion and early termination of PCA.

Variables	Group TBIM (*n* = 102)	Group OBIM (*n* = 102)	*p* Value
Exclusion (No/Yes)	79 (77.5)/23 (22.5)	54 (52.9)/48 (47.1)	<0.001
Causes for Exclusion			
Data loss	13 (12.7)	25 (24.5)	
Early PCA termination	7 (6.9)	13 (12.7)	
Setting error	3 (2.9)	10 (9.8)	
Early PCA termination (No/Yes)	95 (93.1)/7 (6.9)	89 (87.3)/13 (12.7)	0.214
Causes for PCA termination			
Nausea	0 (0.0)	2 (2.0)	
No pain	7 (6.9)	11 (10.8)	

Values are expressed as the number (percentage) of patients. PCA: patient-controlled analgesia, OBIM: optimizing background infusion mode, TBIM: traditional background infusion mode.

**Table 2 medicina-57-00042-t002:** Demographic data and intraoperative variables.

Variables	Group TBIM (*n* = 79)	Group OBIM (*n* = 54)	*p* Value
Age (y	49.7 ± 12.3	49.1 ± 12.7	0.795
Sex (M/F)	39/40	34/20	0.122
Height (cm)	165.5 ± 8.4	166.3 ± 8.2	0.589
Weight (kg)	68.7 ± 13.6	68.4 ± 14.9	0.888
ASA-PS (I/II/III)	39/36/4	31/23/0	0.203
Cumulative remifentanil (µg)	397.9 ± 355.8	369.9 ± 303.9	0.638
Operation time (min)	45.8 ± 45.0	43.9 ± 33.5	0.789
Anesthesia time (min)	59.1 ± 46.5	54.2 ± 33.5	0.503

Values are expressed as the means ± standard deviation or number of patients. ASA-PS: American Society of Anesthesiologists physical status, OBIM: optimizing background infusion mode, TBIM: traditional background infusion mode.

**Table 3 medicina-57-00042-t003:** PCA regimens.

Drugs	Group TBIM (*n* = 79)	Group OBIM (*n* = 54)	*p* Value
Fentanyl (μg)	1195.5 ± 180.7 (*n* =44)	1276.5 ± 172.4 (*n* = 23))	0.081
Sufentanil (μg)	181.8 ± 34.8 (*n* = 35)	169.1 ± 35.9 (*n* = 31)	0.149
Nefopam (mg)	160.0 ± 0.0 (160.0–160.0)	160.0 ± 0.0	1.000
Ramosetron (mg)	1.2 ± 0.0	1.2 ± 0.0	1.000

Values are expressed as means ± standard deviation. PCA: patient-controlled analgesia, OBIM: optimizing background infusion mode, TBIM: traditional background infusion mode.

**Table 4 medicina-57-00042-t004:** Incidence of the requirement for postoperative rescue analgesics and antiemetics.

Variables	Groups	Time				Total
		RR2	6 h	24 h	48 h	
Analgesics	Group TBIM (*n* = 79)	1 (1.3)	14 (17.7)	8 (10.1)	6 (7.6)	23 (29.1)
	Group OBIM (*n* = 54)	1 (1.9)	8 (14.8)	4 (7.4)	1 (1.9)	10 (18.5)
	*p* value	1.000	0.658	0.761	0.240	0.165
Antiemetics	Group TBIM (*n* = 79)	0 (0)	0 (0)	3 (3.8)	0 (0)	3 (3.8)
	Group OBIM (*n* = 54)	0 (0)	2 (3.7)	1 (1.9)	1 (1.9)	3 (5.6)
	*p* value	-	0.163	0.646	0.406	0.686

Values are expressed as the number (percentage) of patients. OBIM: optimizing background infusion mode, RR2: discharge from the recovery room; TBIM: traditional background infusion mode.

**Table 5 medicina-57-00042-t005:** Rescue analgesics.

Variables	Groups		Time			
			RR2	6 h	24 h	48 h
Analgesics	Group TBIM (*n* = 79)	Tramadol	1 (1.3)	14 (17.7)	7 (8.9)	6 (7.6)
		Diclofenac	0 (0)	1 (1.3)	0 (0)	0 (0)
		Fentanyl	0 (0)	0 (0)	1 (1.3)	0 (0)
	Group OBIM (*n* = 54)	Tramadol	1 (1.9)	6 (11.1)	8 (10.1)	1 (1.9)
		Diclofenac	0 (0)	1 (1.9)	4 (7.4)	0 (0)
		Fentanyl	0 (0)	0 (0)	0 (0)	0 (0)
		*p* value	1.000	0.564	0.673	0.240
			Cumulative opioid dose in fentanyl equivalents (μg)		Cumulative diclofenac dose (mg)	
	Group TBIM (*n* = 79)		31.5 (16.3−46.7)		1.1 (−1.1 to 3.4)	
	Group OBIM (*n* = 54)		20.7 (5.2−36.3)		1.7 (−1.7 to 5.0)	
	*p* value		0.339		0.787	
	Mean difference (95% CI)		10.8 (−11.4 to 33.0)		−0.5 (−4.4 to 3.3)	

Values are expressed as the mean (95% confidence interval) or the number (percentage) of patients. OBIM: optimizing background infusion mode, RR2: discharge from the recovery room; TBIM: traditional background infusion mode. Intravenous tramadol (mg) was converted to intravenous fentanyl (μg) according to the suggestion by the following website: https://en.wikipedia.org/wiki/equianalgesic; 1 mg tramadol was equianalgesic to 1 μg fentanyl.

## Data Availability

The data presented in this study are available on request from the corresponding author, through institutional review board, and reviewers. The data are not publicly available due to restrictions of obtaining approval from the IRB for the disclosure of data. If anyone requires our data of this study, please do not hesitate to contact the corresponding author.

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
