# Peer review of "The Optimizing Background Infusion Mode Decreases Intravenous Patient-Controlled Analgesic Volume and Opioid Consumption Compared to Fixed-Rate Background Infusion in Patients Undergoing Laparoscopic Cholecystectomy: A Prospective, Randomized, Controlled, Double-Blind Study"

_medicina, 2021, doi:10.3390/medicina57010042_

Round 1

Reviewer 1 Report

Thank you for permitting to review here are my suggestions 

abstract: traditional pca  don't have always infusion ,  indeed most of the time there is no infusion  in postoperative PCA, this sentence must be improved in order not to confuse the reader

Introduction : the primary objective is not cited in the introduction , the first time  that we read primary objevtives is in the results section  

methods: 

it is better to state post anesthetic care unit , instead of recovery room

results: 

I am surprised that pca and opioid demand are so high , average lap chole procedure requires 25 mg of morphine PCA0

Was there any local anesthesia infiltration , if not it should be clearly stated

Discussion 

the authors should discuss the issue why it is better to inject more  opioid early in the course of postoperative pain and less later , since it appears this is the major outcome 

Author Response

Response to Reviewer 1 Comments

  • We used the line number in the manuscript, which maintained the "Track Changes" function in Microsoft Word.

Point 1: Abstract: traditional PCA don’t have always infusion, indeed most of the time there is no infusion in postoperative PCA, and this sentence must be improved in order not to confuse the reader.

Response 1: Thank you for pointing this out. We agree with this comment. According to the reviewer's comment, we revised it as shown in the sentence below (in lines 16-18) :

“The fixed-rate continuous background infusion mode with bolus dosing is one of common modalities for intravenous patient-controlled analgesia (PCA).”

Point 2: Introduction: the primary objective is not cited in the introduction, the first time that we read primary objectives is in the results section.

Response 2: Thank you for pointing this out. We agree with this comment. According to the reviewer's comment, the sentence for primary outcome described in Materials and Methods was described in the section of the introduction, as shown in the sentence below (in lines 71-72):

“Therefore, we investigated the efficacy of PCA using bolus dosing and either the OBIM or the TBIM for postoperative analgesia in patients undergoing laparoscopic cholecystectomy. The primary outcome of this study was NRS score for pain achieving a medium effect size (0.5) at the 6th postoperative hours.”

Point 3: Methods: it is better to state post anesthetic care unit, instead of recovery room.

Response 3: Thank you for pointing this out. We agree with this comment. Actually, the system for postanesthetic patient care varies from country to country or from hospital to hospital. Hospitals in some countries are using a system to manage patients once again in the recovery room before being transferred to the ward after sufficient recovery from the postanesthetic care units. However, most hospitals in Korea, including our own, are applying a system that operates only recovery rooms for postanesthetic care. Therefore, we describe it using a recovery room instead of a postanesthetic care unit. Please consider this point.

Point 4: Results: I am surprised that PCA and opioid demand are so high, average lap chole procedure requires 25 mg of morphine PCA0. Was there any local anesthesia infiltration, if not it should be clearly stated. 

Response 4: Thank you for pointing this out. In Korea, most surgeons are not interested in actively controlling postoperative pain, considering that the postoperative pain is very low after the laparoscopic surgery, and have chosen one of the modalities for postoperative analgesia at the request of the patient preoperatively or postoperatively. In our hospital, we also do not implement additional analgesic modalities such as wound infiltration or regimental analgesia with local anesthetic other than intravenous PCA as routine at the end of surgery. Therefore, all patients enrolled in this study were not given additional analgesic modalities at the end of surgery.

We revised it as shown in the sentence below (in lines 130-131):

“At the end of surgery, the patients, were not received any wound anesthetic infiltration with local anesthetics or any regional analgesia.” The patients were transferred to the recovery room (RR) after the complete reversal of rocuronium-induced neuromuscular paralysis and being fully awake.

Point 5: Discussion: the authors should discuss the issue why it is better to inject more opioid early in the course of postoperative pain and less later, since it appears this is the major outcome.

Response 5: Thank you for pointing this out. We agree with this comment. According to the reviewer's comment, we further described the benefits of using OBIM in a new paragraph of discussion, as shown in the sentence below (in lines 334-349):

“Most patients experience a biphasic pattern of pain intensity that requires more analgesics immediately after surgery and then requires less analgesics [6,7]. The fixed background infusion rate (TBIM) combined with bolus dosing is not enough to overcome the early postoperative pain, because the lockout interval limits the rescue analgesia infusion via PCA despite of frequent bolus demands [6,7,18]. However, OBIM with bolus dosing has a benefit of providing more effective postoperative pain and reducing total infused volume than TBIM with bolus dosing [10,18], because OBIM increase the background infusion rate as the increased bolus demands immediately after surgery, and then decrease background infusion rate according to the absence of bolus dosing requirement [10]. We have commonly adopted the PCA using bolus dosing combined with TBIM. However, TBIM showed the higher incidence of insufficient postoperative analgesia and the higher postoperative pain, resulted in increasing the total opioid consumption with higher postoperative bolus requirements compared with OBIM [18]. Our study also showed that OBIM reduced the total infused volume (total opioid consumption) even though there was the non-significant reduced postoperative NRS in improving postoperative analgesia. So, PCA using OBIM combined with bolus dosing can provide more sufficient postoperative analgesia and reduce opioid consumption more by decreasing total infused volume.”

  • We have revised some words and sentences in the revision process according to comments of reviewers. We have highlighted the changes within the manuscript using the "Track Changes" function in Microsoft Word.

Reviewer 2 Report

The authors present a clinical study on the novel approach of intravenous opioid patient-controlled analgesia (PCA) use. In general the article is well written, nevertheless, I suggest to make following changes in order to make the conlusions stronger.

These are my comments:

Major revisions:

Doses of fentanyl and sufentanyl instead of volume should be reported as two different opioids were used; and although not significant but slight differences exist in prescribed PCA regimens between the groups (Table 3), this may have effect on total cumulative opioid doses used. 

There is only very limited data redarding adverse effects in the Results section. Please exclude the adverse effects from the hypothesis.

Other:

Methodology

  1. Blinding is questioned as the blinding was not secured until the statistical analysis or at least until the the end of involvement of the patient in the trial. Anesthesiologist, who collected the trial data could find the group assignment on the anesthesia record. Who followed up  (pain scores, PONV) the patients? This should be clarified.
  2. Patients were allowed additional analgesics and opioids on the ward - the cumulative doses of additional analgesics, including opioids and tramadol should be reported in results section.
  3. on what the selected PCA setting were based (please explain or provide reference)
  4. How PONV was rated?
  5. Primary and secondary outcomes should be clearly defined as per CONSORT checklist
  6. I can not report on statistical analysis.

Results

  1. Lines 169-170 - please provide percent  
  2. Please, correct the presentation of results according to the requirements of CONSORT checklist (Table 2)
  3. Table 3 - values in microg/kg of opioids does not correspond to values in methodology ( they are half of values given in methodology section)? Why? Please explain, discuss or correct methodology.
  4. P values on every time point in the graphs 2, 3, 4 and 5 are excessive 
  5. Line 228 - change word "final" to "total"
  6. Lines 229-230 - please provide the mean difference (95% CI) as per CONSORT requirements for the difference in the total cumulative dose.
  7. TABLE 4 , please provide the cumulative doses of additional opioids and other analgesics as this would strenghen the conclusion that comparable analgesia was achieved with lower doses of fentanyl/or sufentanyl and additional analgesics with the use of OBIM.

DISCUSSION

16. I would suggest to start the discussion with the statement that OBIM offers adequate response to bifasic pain pattern after laparoscopic surgery compared to traditional PCA.

17. Line 255 - change "final" to 'total"

18. Lines 285-300 - no data regarding PONV are presented in the results section. Correct results or discussion accordingly.

Other limitations

19. Not Assessing  sedation? and respiartory function? - should be discussed and included into the limitations.

20. Multimodal pain management which is currently recommended was not routinely applied - this should be discussed in the limitations section.

CONCLUSION

21. Again,  I suggest to emphasize in the conclusion the adequate response to bifasic pain pattern with OBIM after surgery. As I am not sure that the cumulative dose (but not volume) of opioid would be reduced with OBIM (therefore results presenting opioid dose instead of volume should be presented). Nevertheless, this method may offer advantages over traditional PCA.  In case the doses may not differ, this study is still important as it describes the novel approach to treatment with PCA, which is not inferior, but may be superior to traditional PCA. 

Author Response

Response to Reviewer 2 Comments

  • We used the line number in the manuscript, which maintained the "Track Changes" function in Microsoft Word.

Major revisions:

Point 1: Doses of fentanyl and sufentanil instead of volume should be reported as two different opioids were used; and although not significant but slight differences exist in prescribed PCA regimens between the groups (Table 3), this may have effect on total cumulative opioid doses used.

Response 1: Thank you for pointing this out. We analysed differences of the cumulative and per-interval infused fentanyl doses between two groups after we converted sufentanil doses to equianalgesic doses of fentanyl instead of using of individual doses of fentanyl and sufentanil. We further described results of time-sequential changes of cumulative and per-interval infused fentanyl doses in the results (3.5. Infused PCA Volumes and Infused Opioid Doses) as follows, and added the figure 6 (in lines 263-281).  

“In addition, we analyzed differences of the cumulative and per-interval infused fentanyl doses between two groups after the sufentanil doses were converted to equianalgesic dose of fentanyl. The cumulative infused opioid dose was significantly different throughout the postoperative period (P < 0.001) and at each measured interval (P ≤ 0.006) except at the 18th, 24th and 30th postoperative hours (Figure 6a). It was higher in group OBIM than in group TBIM until the 12th postoperative hour, and lower from the 38th to the 48th postoperative hour (Figure 6a). The total cumulative opioid dose was lower in group OBIM [714.1 (647.4−780.9) μg of fentanyl] than in group TBIM [963.7 (870.5−1056.9 μg of fentanyl] (mean difference (96% CI): 58.0 (-133.5 to 365.6), P < 0.001, Figure 6a). The per-interval infused fentanyl dose was significantly different between groups throughout the postoperative period (P < 0.001, Figure 6b). It was higher in group OBIM than in group TBIM until the 6th postoperative hour, and lower from the 18th to the 48th hours (P ≤ 0.008, Figure 6b).”

Point 2: There is only very limited data regarding adverse effects in the Results section. Please exclude the adverse effects from the hypothesis.

Response 2: Thank you for pointing this out. We agree with this comment. We excluded the adverse effects from the hypothesis as shown in the sentence below (in lines 67-68):

 “We hypothesized that the OBIM would provide better postoperative analgesia and a lower cumulative opioid consumption, with the traditional (fixed-rate) background infusion mode (TBIM).”

Other:

Methodology

Point 3: Blinding is questioned as the blinding was not secured until the statistical analysis or at least until the end of involvement of the patient in the trial. Anesthesiologist, who collected the trial data could find the group assignment on the anesthesia record. Who followed up (pain scores, PONV) the patients? This should be clarified.

Response 3: Thank you for pointing this out. An anesthesiologist who collected the trial data from patients collated them from medical records and PCA devices at least 48 hours after surgery. During this period, this anesthesiologist did not participate in any assessment and intervention of the patients. Well-trained ward nurses conducted assessment of pain and PONV and recorded them in the medical record. We further described it as follows (in lines 103-108).

“For blinding, RP recorded the drug assignment on anesthesia record paper after the anesthesia was completely finished, and RA finally collated the data of patient medical records and that generated in the trial at least after 48th postoperative hours. The nurses in the RR or ward recorded the postoperative pain and PONV with NRS. They were non-investigating nurse, and trained in the hospital to assess pain intensities and PONV with NRS. Neither RA nor RP participated in the statistical analysis.”

Point 4: Patients were allowed additional analgesics and opioids on the ward - the cumulative doses of additional analgesics, including opioids and tramadol should be reported in results section.

Response 4: Thank you for pointing this out. The postoperative rescue analgesics were tramadol, diclofenac, and fentanyl. After we converted each rescue analgesics doses to fentanyl dose with each conversion ratio with fentanyl, we analyzed the total cumulative equianalgesic doses with fentanyl. We further described it as follows, and added the Table 5 (in lines 286-290, in lines 295-301).

“The postoperative rescue analgesics were tramadol, diclofenac, and fentanyl, which were not significantly different between the groups throughout the recovery period (p ≥ 0.05, Table 5). The total cumulative equianalgesic doses with fentanyl converted from each rescue analgesic were not significantly different between the groups [mean 32.2 μg and 21.8 μg in groups TBIM and OBIM, respectively, and mean difference (96% CI): 10.4 (-12.2 to 33.1)] (P = 0.364).”

Point 5: on what the selected PCA setting were based (please explain or provide reference)

Response 5: Thank you for pointing this out. When we designed this study, there were no papers on the basis of the OBIM setup. So, comprehensively reflecting the opinions of manufacturers and researchers due to lack of evidence for the OBIM settings, we have finalized these settings that can ensure patient safety and provide effective analgesia. According to the reviewer's comment, the authors wrote the percentage as shown in the sentence below (in lines 127-129):

“Comprehensively reflecting the opinions of manufacturers and researchers due to lack of evidence for the OBIM settings, we have finalized these settings that can ensure patient safety and provide effective analgesia.”

Point 6: How PONV was rated?

Response 6: Thank you for pointing this out. We used NRS for evaluation of PONV. This was described in materials and methods as follows (in lines 136-141).

“We also allowed the intravenous injection of opioids, nonsteroidal anti-inflammatory drugs, or tramadol as a rescue analgesic to treat pain of >4 points on the NRS in the ward. We treated postoperative nausea and vomiting (PONV) of >4 points on the NRS with intravenous injection of metoclopramide (10 mg). Our research staff decided whether to stop the PCA device or change its setting based on severity of signs and symptoms, and we excluded such cases from the final statistical analysis.”

Point 7: Primary and secondary outcomes should be clearly defined as per CONSORT checklist.

Response 7: Thank you for pointing this out. The primary outcome in this study was postoperative pain, which evaluated with NRS at 6th postoperative hours. In addition, we assumed that the effect size would be medium size (0.5). So, we described the definition of primary outcome, including how and when they were assessed, at the end of the introduction as follow (in lines 71-72):

“The primary outcome of this study was NRS score for pain achieving a medium effect size (0.5) at the 6th postoperative hours.”

Point 8: I cannot report on statistical analysis.

Response 8: There is nothing to answer.

Results

Point 9: Lines 169-170 - please provide percent.

Response 9: Thank you for pointing this out. According to the reviewer's comment, the authors wrote the percentage as shown in the sentence below (in lines 179-180):

 “The causes of exclusion were data loss during collection in 38 patients (18.6%), early PCA termination in 20 patients (9.8%), and device setting errors in 13 patients (6.4%) (Table 1).”

Point 10: Please, correct the presentation of results according to the requirements of CONSORT checklist (Table 2)

Response 10: Thank you for pointing this out. After checking the requirements of CONSORT checklist, we revised the values described in Table 2 and Table 3 according to the requirements of CONSORT checklist. We re-described then with means ± SD instead of means and 95% confidence intervals (in lines 196-203).

Point 11: Table 3 - values in microg/kg of opioids does not correspond to values in methodology (they are half of values given in methodology section)? Why? Please explain, discuss or correct methodology.

Response 11: Thank you for pointing this out. We agree with this comment. We found that the doses of fentanyl or sufentanil were mistyped to ‘0 μg’ instead of blank in ‘case report form’ in patients who did not receive any opioid. So, we reanalyzed the data for each opioid dose by classifying patients who received fentanyl and patients who received sufentanil in each group, and revised the results in table 3 (in lines 200-203).

Point 12: P values on every time point in the graphs 2, 3, 4 and 5 are excessive

Response 12: Thank you for pointing this out. We agree with this comment. According to the reviewer's comment, we deleted the P values shown above the values in Figures 2, 3, 4, and 5, and marked P < 0.05 with '*' for P < 0.001 with '†'. We also described this in each figure legends.

Point 13: Line 228 - change word "final" to "total"

Response 13: Thank you for pointing this out. According to the reviewer's comment, we revised it (in lines 245-247).

“The total cumulative infused volume was lower in group OBIM [84.0 (78.9−89.1) mL) than in group TBIM [102 (97.8−106.0) mL] (mean difference (96% CI): 17.9 (11.6 to 24.2), P < 0.001, Figure 5a).”

Point 14: Lines 229-230 - please provide the mean difference (95% CI) as per CONSORT requirements for the difference in the total cumulative dose.

Response 14: Thank you for pointing this out. According to the reviewer's comment, we further described the mean difference (95% CI) as per CONSORT requirements for the difference in the total cumulative dose in a new paragraph of results (3.5. Infused PCA Volumes and Infused Opioid Doses) as shown in the sentence below (in lines 263-273):

“In addition, we analyzed differences of the cumulative and per-interval infused fentanyl doses between two groups after the sufentanil doses were converted to equianalgesic dose of fentanyl. The cumulative infused opioid dose was significantly different throughout the postoperative period (P < 0.001) and at each measured interval (P ≤ 0.006) except at the 18th, 24th and 30th postoperative hours (Figure 6a). It was higher in group OBIM than in group TBIM until the 12th postoperative hour, and lower from the 38th to the 48th postoperative hour (Figure 6a). The total cumulative opioid dose was lower in group OBIM [714.1 (647.4−780.9) μg of fentanyl] than in group TBIM [963.7 (870.5−1056.9 μg of fentanyl] (mean difference (96% CI): 58.0 (-133.5 to 365.6), P < 0.001, Figure 6a). The per-interval infused fentanyl dose was significantly different between groups throughout the postoperative period (P < 0.001, Figure 6b). It was higher in group OBIM than in group TBIM until the 6th postoperative hour, and lower from the 18th to the 48th hours (P ≤ 0.008, Figure 6b).”

In addition, we further described the mean difference (95% CI) as per CONSORT requirements for the difference in the total cumulative infused PCA volume in the results (3.5. Infused PCA Volumes and Infused Opioid Doses) as shown in the sentence below (in lines 245-247):

“The total cumulative infused volume was lower in group OBIM [84.0 (78.9−89.1) mL) than in group TBIM [102 (97.8−106.0) mL] (mean difference (96% CI): 17.9 (11.6 to 24.2), P < 0.001, Figure 5a).

Point 15: TABLE 4, please provide the cumulative doses of additional opioids and other analgesics as this would strengthen the conclusion that comparable analgesia was achieved with lower doses of fentanyl/or sufentanil and additional analgesics with the use of OBIM.

Response 15: Thank you for pointing this out. The postoperative rescue analgesics were tramadol, diclofenac, and fentanyl. After we converted each rescue analgesics doses to fentanyl dose with each conversion ratio with fentanyl, we analyzed the total cumulative equianalgesic doses with fentanyl. We further described it as follows (in lines 286-290), and added the Table 5 (in lines 295-301).

“The postoperative rescue analgesics were tramadol, diclofenac, and fentanyl, which were not significantly different between the groups throughout the recovery period (p ≥ 0.05, Table 5). The total cumulative equianalgesic doses with fentanyl converted from each rescue analgesic were not significantly different between the groups [mean 32.2 μg and 21.8 μg in groups TBIM and OBIM, respectively, and mean difference (96% CI): 10.4 (-12.2 to 33.1)] (P = 0.364).”

DISCUSSION

Point 16: I would suggest to start the discussion with the statement that OBIM offers adequate response to biphasic pain pattern after laparoscopic surgery compared to traditional PCA.

Response 16: Thank you for pointing this out. According to the reviewer's comment, we added new sentence in first paragraph of discussion as follow (in lines 307-309):

“This prospective, double-blind, randomized, controlled study revealed that the NRS score and the bolus demand count did not differ between groups throughout the recovery period. Patients in group OBIM exhibited a higher background infusion rate before the 12th postoperative hour, and a lower rate from the 12th to the 48th postoperative hours, compared with those in group TBIM. OBIM offered adequate response to biphasic pain pattern after laparoscopic surgery compared to TBIM. The total cumulative infused opioid dose and PCA volume were lower in group OBIM than in group TBIM.”

Point 17: Line 255 - change "final" to 'total"

Response 17: Thank you for pointing this out. According to the reviewer's comment, we revised it as follow (in line 309):

“The total cumulative infused opioid dose and PCA volume were lower in group OBIM than in group TBIM.”

Point 18: Lines 285-300 - no data regarding PONV are presented in the results section. Correct results or discussion accordingly.

Response 18: Thank you for pointing this out. Even though the nurses recorded the NRS for PONV, we just collated PONV requiring antiemetics during postoperative period and showed the number of patients in table 4. Table 4 showed the higher incidence of PONV requiring antiemetics at 6th postoperative hours in group OBIM, and at 24th postoperative hours in group TBIM. So, we described it based on results in first submitted manuscript as follow (in lines 355-361):

“This study revealed that postoperative nausea requiring antiemetics mainly occurred before the 6th postoperative hour in the OBIM group (3.7%), and after the 6th postoperative hour in the TBIM group (3.8%) (Table 4). No other adverse effects were observed.

Lee at al. [10] documented that the overall incidence of PONV requiring antiemetics was lower in the OBIM group (18%) compared with the TBIM group (33%), whereas, in this study, it was higher in the OBIM group (5.6%) compared with the TBIM group (3.8%) (Table 4).”

Please take this into consideration.

Other limitations

Point 19: Not assessing sedation? And respiratory function? - should be discussed and included into the limitations.

Response 19: Thank you for pointing this out. After applying PCA, we collected data on side effects such as sedation and respiratory suppression. However, neither of the groups of OBIM nor TBIM have experienced these side effects. This was described as the last sentence of the fourth paragraph of discussion as follow (in line 357):

“This study revealed that postoperative nausea requiring antiemetics mainly occurred before the 6th postoperative hour in the OBIM group (3.7%), and after the 6th postoperative hour in the TBIM group (3.8%) (Table 4). No other adverse effects were observed.”

So, we did not discussed them, and did not included into the limitation. Please take this into consideration.

Point 20: Multimodal pain management which is currently recommended was not routinely applied - this should be discussed in the limitations section.

Response 20: Thank you for pointing this out. We agree with this comment that multimodal pain management protocols have consistently been demonstrated to allow for improved pain control with less reliance on opioids. Actually, into this study we did not adopted multimodal pain management protocols, such as preemptive analgesia, NSAID, gabapentinoids, acetaminophen, muscle relaxants, ketamine, neuroaxial blockade, and local infiltrative anesthetic. So, we included it into the limitation as follow (in lines 383-387):

“Another limitation of this study is that we did not adopted multimodal pain management protocols, such as preemptive analgesia, NSAID, gabapentinoids, acetaminophen, muscle relaxants, ketamine, neuroaxial blockade, and local infiltrative anesthetic. [19]. Therefore, we cannot conclude that OBIM will be able to provide more effective postoperative analgesia in the context of applying the multimodal pain management protocol with PCA.”

CONCLUSION

Point 21: Again, I suggest to emphasize in the conclusion the adequate response to biphasic pain pattern with OBIM after surgery. As I am not sure that the cumulative dose (but not volume) of opioid would be reduced with OBIM (therefore results presenting opioid dose instead of volume should be presented). Nevertheless, this method may offer advantages over traditional PCA.  In case the doses may not differ, this study is still important as it describes the novel approach to treatment with PCA, which is not inferior, but may be superior to traditional PCA.

Response 21: Thank you for pointing this out. According to the reviewer's comment, we revised it as follow (in lines 389-391):

“In conclusion, the OBIM of PCA is useful in that it reduces the cumulative opioid dose and PCA volume by responding more effectively to postoperative pain compared to the TBIM, while yielding comparable postoperative analgesia and bolus demand in patients undergoing laparoscopic cholecystectomy. In addition, further studies are required to determine the efficacy of the OBIM of PCA considering different types of surgery and degrees of postoperative pain.”

  • We have revised some words and sentences in the revision process according to comments of reviewers. We have highlighted the changes within the manuscript using the "Track Changes" function in Microsoft Word.

Reviewer 3 Report

Please, update the references, so as to include information from the last 5 years.

It is a very interesting research study.

Author Response

Response to Reviewer 3 Comments

  • We used the line number in the manuscript, which maintained the "Track Changes" function in Microsoft Word.

Point 1: Please, update the references, so as to include information from the last 5 years. It is a very interesting research study.

Response 1: Thank you for pointing this out. According to the reviewer’s comment, we tried to update the references, so as to include information from the last 5 years. However, we did not find another recent references to replace the references cited in this manuscript, because our study was one of the novel methods for PCA using OBIM. Instead, we quoted several more new references during the review according to comments from other reviewers.

Please take this into consideration.

  • We have revised some words and sentences in the revision process according to comments of reviewers. We have highlighted the changes within the manuscript using the "Track Changes" function in Microsoft Word.

Round 2

Reviewer 1 Report

The authors had adequately responded to queries, moderate english modifications will improve the final version of the manuscript 

Author Response

Response to Reviewer 1 Comments

  • We used the line number in the manuscript, which maintained the "Track Changes" function in Microsoft Word.

Point 1: The authors had adequately responded to queries, moderate english modifications will improve the final version of the manuscript.

Response 1: Thank you for your comments and for highlighting this issue. We have had the manuscript reviewed again by a native English speaker and changes within the manuscript have been highlighted using the "Track Changes" function in Microsoft Word.

Reviewer 2 Report

Revised version Review 12.29.

Please, make these additional changes:

  1. Figure 4a is not cited in the text. Please correct..
  2. Lines 269-271: The numbers in this sentence does not make sense - either p or mean difference (95% CI) are wrong, please correct: . The total cumulative opioid dose was  lower in group OBIM [714.1 (647.4−780.9) μg of fentanyl] than in group TBIM [963.7 (870.5−1056.9 μg  of fentanyl] (mean difference (96% CI): 58.0 (-133.5 to 365.6), P < 0.001,
  3. Please change Table, Figure and axes names where necessary to opioid dose in fentanyl equivalents; as well as in all corresponding text. 
  4. Table 5: There is no sense in converting diclofenac to Fentanyl - please provide diclofenac dose not converted. Provide cumulative opioid dose in fentanyl equivalents only for opioids (fentanyl and tramadol). Correct all corresponding text accordingly.
  5. English language needs extensive correction in some newly included places.
  6. Lastly, I would suggest to shorten the Title as it is very heavy...

Author Response

Response to Reviewer 2 Comments

  • We used the line number in the manuscript, which maintained the "Track Changes" function in Microsoft Word.

Point 1: Figure 4a is not cited in the text. Please correct.

Response 1: Thank you for your helpful comments. We have cited figure 4a in the text as follows (line 231):

“The background infusion rate was significantly different between groups throughout the postoperative period (P < 0.001, Figures 4a and 4b).”

Point 2: Lines 269-271: The numbers in this sentence does not make sense - either p or mean difference (95% CI) are wrong, please correct: The total cumulative opioid dose was lower in group OBIM [714.1 (647.4−780.9) μg of fentanyl] than in group TBIM [963.7 (870.5−1056.9 μg of fentanyl] (mean difference (96% CI): 58.0 (-133.5 to 365.6), P < 0.001,

Response 2: We agree with your comment. We identified the numerical errors in this sentence and corrected them as follows (in lines 270-272):

“The total cumulative opioid dose was lower in group OBIM (714.1 [647.4−780.9] μg) than in group TBIM (963.7 [870.5−1056.9] μg; mean difference [95% CI]: 249.6 [133.5 to 365.6]; P < 0.001, Figure 6a).”

We also changed ‘fentanyl dose’ to ‘opioid dose in fentanyl equivalents’ according to the reviewer's comment (point 3 below), as follows (in lines 264-286, in figures 6a and 6b):

“In addition, we analyzed differences of the cumulative and per-interval infused opioid doses between the two groups after the sufentanil doses were converted to opioid doses in fentanyl equivalents. The cumulative infused opioid dose was significantly different throughout the postoperative period (P < 0.001) and at each measured interval (P ≤ 0.006) except at the 18th, 24th and 30th postoperative hours (Figure 6a). It was higher in group OBIM than in group TBIM until the 12th postoperative hour and lower from the 38th to the 48th postoperative hour (Figure 6a). The total cumulative opioid dose was lower in group OBIM (714.1 [647.4−780.9] μg) than in group TBIM (963.7 [870.5−1056.9] μg; mean difference [95% CI]: 249.6 [133.5 to 365.6]; P < 0.001, Figure 6a). The per-interval infused opioid dose was significantly different between groups throughout the postoperative period (P < 0.001, Figure 6b). It was higher in group OBIM than in group TBIM until the 6th postoperative hour and lower from the 18th to the 48th hours (P ≤ 0.008, Figure 6b).”

Point 3: Please change Table, Figure and axes names where necessary to opioid dose in fentanyl equivalents; as well as in all corresponding text.

Response 3: Thank you for this helpful suggestion. We have now changed ‘fentanyl dose’ to ‘opioid dose in fentanyl equivalents’ as per your suggestion (in lines 264-286, in figures 6a and 6b):

“In addition, we analyzed differences of the cumulative and per-interval infused opioid doses between the two groups after the sufentanil doses were converted to opioid doses in fentanyl equivalents. The cumulative infused opioid dose was significantly different throughout the postoperative period (P < 0.001) and at each measured interval (P ≤ 0.006) except at the 18th, 24th and 30th postoperative hours (Figure 6a). It was higher in group OBIM than in group TBIM until the 12th postoperative hour and lower from the 38th to the 48th postoperative hour (Figure 6a). The total cumulative opioid dose was lower in group OBIM (714.1 [647.4−780.9] μg) than in group TBIM (963.7 [870.5−1056.9] μg; mean difference [95% CI]: 249.6 [133.5 to 365.6]; P < 0.001, Figure 6a). The per-interval infused opioid dose was significantly different between groups throughout the postoperative period (P < 0.001, Figure 6b). It was higher in group OBIM than in group TBIM until the 6th postoperative hour and lower from the 18th to the 48th hours (P ≤ 0.008, Figure 6b).”

We have also changed ‘fentanyl dose’ to ‘opioid dose in fentanyl equivalents’, as follows (in lines 293-299, lines 305-311, and Table 5):

“The total cumulative opioid dose after the tramadol doses were converted to opioid doses in fentanyl equivalents was not significantly different between the groups (mean: 31.5 μg and 20.7 μg in groups TBIM and OBIM, respectively, and mean difference [95% CI]: 10.8 [-11.4 to 33.0]; P = 0.339, Table 5). The total cumulative diclofenac dose was not significantly different between the groups [mean: 1.1 mg and 1.7 mg in groups TBIM and OBIM, respectively, and mean difference [95% CI]: -0.5 [-4.4 to 3.3]; P = 0.787, Table 5).”

Point 4: There is no sense in converting diclofenac to Fentanyl - please provide diclofenac dose not converted. Provide cumulative opioid dose in fentanyl equivalents only for opioids (fentanyl and tramadol). Correct all corresponding text accordingly.

Response 4: As per your comment, we have provided the cumulative opioid dose in fentanyl equivalents only for opioids (fentanyl and tramadol), and we have provided the unconverted diclofenac dose. We corrected Table 5 and further described this as follows (in lines 293-299, in lines 305-311):

“The total cumulative opioid dose after the tramadol doses were converted to opioid doses in fentanyl equivalents was not significantly different between the groups (mean: 31.5 μg and 20.7 μg in groups TBIM and OBIM, respectively, and mean difference [95% CI]: 10.8 [-11.4 to 33.0]; P = 0.339, Table 5). The total cumulative diclofenac dose was not significantly different between the groups [mean: 1.1 mg and 1.7 mg in groups TBIM and OBIM, respectively, and mean difference [95% CI]: -0.5 [-4.4 to 3.3]; P = 0.787, Table 5).”

Point 5: English language needs extensive correction in some newly included places.

Response 5: Thank you for this comment. As per our response to Reviewer 1, we have had the manuscript reviewed again by a native English speaker and changes within the manuscript have been highlighted using the "Track Changes" function in Microsoft Word.

Point 6: Lastly, I would suggest to shorten the Title as it is very heavy...

Response 6: We have discussed this issue and, in accordance with the principles of PICO, we feel that it is important to include all the information currently contained in the title. As a result, we have decided not to modify the title. We do hope that you understand our position regarding this issue.

“The optimizing background infusion mode decreases intravenous patient-controlled analgesic volume and opioid consumption compared to fixed-rate background infusion in patients undergoing laparoscopic cholecystectomy: a prospective, randomized, controlled, double-blind study”
